# High-Resolution Crack Localization Approach Based on Diffraction Wave

**DOI:** 10.3390/s19081951

**Published:** 2019-04-25

**Authors:** Weilei Mu, Jiangang Sun, Guijie Liu, Shuqing Wang

**Affiliations:** Shandong Provincial Key Laboratory of Ocean Engineering, Ocean University of China, Qingdao 266100, China; tblueapple@126.com (W.M.); sjg19940721@126.com (J.S.)

**Keywords:** ultrasonic guided wave, dispersion, high resolution detection, damage imaging

## Abstract

The delay-and-sum imaging algorithm is a promising crack localization approach for crack detection and monitoring of key structural regions. Most studies successfully offer a hole-like damage position. However, cracks are more common than hole-like damages in a structure. To solve this issue, this paper presents a crack localization approach, based on diffraction wave theory, which is capable of imaging crack endpoints. The guided wave propagated to the crack endpoints and transformed into a diffraction wave. A line sensor array was used to record the diffraction waveform. Then, dispersion compensation was applied to shorten the dispersive wave packets and separate the overlapping wave packets. Subsequently, half-wave compensation was executed to improve the localization accuracy. Finally, the effectiveness of this high-resolution crack localization method was validated by an experimental example.

## 1. Introduction

During service, offshore platforms are affected by wind, waves, sea current, and other alternating loads, which can lead to fatigue cracks in key welding regions, such as welds of arc-soft-toe bracket (ASTB) joints. Moreover, corrosion and iceberg impact also produce fatigue cracks in critical areas, even though the welding quality is good. The crack appears when the bracket toe is not soft enough to release the stress concentration. Therefore, cracks often appear at the end of the toe of the elbow and the direction is approximately perpendicular to the weld [1]. Conventional annual inspection or docking detection of offshore platforms is insensitive to micro-fatigue cracks, leading to missed detection of fatigue cracks. Therefore, developing a structural health monitoring technique for offshore platforms is highly required [2].

With advantages of long propagation distance, low cost, and good sensitivity to various defects [3], ultrasonic guided wave is considered a promising non-destructive testing approach for monitoring and detecting damage [4]. Mofakhami and Boller [5] used Lamb waves to detect both-sided cracks on the edge of a circular hole. A relatively original approach to the problem of crack detection in a rivet joint is presented by Liu et al. [6]. However, guided wave signals are difficult to analyze due to their complex characteristics, such as dispersion, multi-mode character, and mode conversion. Most importantly, wave packets expand in the time domain as they propagate. As a result, damage with a high resolution is considerably difficult to localize [7].

Wang proposed a time reversal approach for compressing extended wave packets to reduce the deformation induced by the effect of dispersion characteristics, but it cannot eliminate the effect [8]. By using prior knowledge on the dispersion curve of a certain guided wave mode, Wilcox developed a dispersion compensation technique by mapping signals from the time domain to the distance domain [9,10]. Sicard proposed a back-propagation compensation technique, which does not require the propagation distance [11]. This method has been widely used since it was proposed because the propagation distance is always unknown at the beginning. On this basis, Liu proposed a linear mapping technique for the dispersion removal of guided waves; it transformed the original priori known dispersion relation into a linear dispersive relation [12]. Inspired by the time–distance domain mapping method introduced by Wilcox, Zeng designed a high-resolution damage imaging method [13]. With this method, Zeng detected two adjacent through-thickness holes by compensating the reflected wave signals. However, most researchers have attempted to validate the proposed methods through experiments with through-thickness holes, whereas actual cracks are rarely shaped like a circle in structural weld regions. Nonlinear wave features are extracted using networked, miniaturized piezoelectric wafers and reverted to the time domain for damage localization. The proposed approach circumvents the deficiencies of using Lamb wave features for evaluating under-sized damage by Hong [14]. The structural health monitoring system based on the active pitch-catch measurement technique was proposed. A comparison of the intact and defected structures is used by the damage detection algorithm [15]. The applied method is based on a visualization of sensing paths with an assigned value of correlation coefficients, computed for reference and actual signals [16]. For this reason, a high-resolution imaging method for fatigue cracks is proposed in the present study.

This study aims to improve the resolution and localization accuracy of fatigue cracks. To solve the problem of low resolution, a time–distance mapping compensation method was designed to separate overlapped wave packets. Then, the delay-and-sum algorithm was used to process the diffraction signals of array sensors, and half-wave compensation was adopted to improve the localization accuracy. Finally, a compensation simulation and location experiment were executed to validate the proposed method.

## 2. Dispersion Compensation Method

### 2.1. Cause of Dispersion

Generally, the structural elements connected by ASTB joints in an offshore platform are plate-like components or pipes with large radii, as shown in Figure 1. In a relatively small region, the pipe could be regarded as a plate-like component approximately. When the ultrasonic guided wave propagates in plate-like structures, the waveform will be deformed and expanded in the time and space domain, because the phase velocity of each frequency component in the signal is different. This phenomenon is called dispersion of waves.

In a pitch-catch scenario, a narrow band signal of f(t) is commonly used as an excitation signal, which can excite the ultrasonic guided wave in a plate. After propagating over a certain distance x, each frequency component of the guided wave will arrive at x with a time delay of $t_{i}=x/c_{p}(\varpi )$, where $c_{p}(\varpi )$ is the phase velocity of a narrow band signal with a center frequency ϖ. The waveform after propagation can be expressed as follows [17]:(1)$$\begin{array}{l} u(x,t)=\int_{-\infty }^{+\infty }A(\varpi )F(\varpi )e^{-i\varpi x/c_{p}}e^{i\varpi t}d\varpi \\ \;\;\;\;\;\;\;\;\;\;\;=\int_{-\infty }^{+\infty }A(\varpi )F(\varpi )e^{-ik(\varpi )x}e^{i\varpi t}d\varpi \end{array}$$
where k(ϖ) is the wave number of the signal, $k(\varpi )=\varpi /c_{p}(\varpi )$, F(ϖ) denotes the Fourier transformation of f(t), and A(ϖ) indicates the amplitude attenuation coefficient.

If the receiver is located at a point with a distance L from the exciter, then the predicted signals are as follows [18]:(2)$$u(t)=\int_{-\infty }^{+\infty }A(\varpi )F(\varpi )e^{-ik(\varpi )L}e^{i\varpi t}d\varpi$$

Practically, the relationship between phase velocity $c_{p}(\varpi )$ and frequency ϖ is often nonlinear. Thus, the relationship between wavenumber k(ϖ) and frequency ϖ is also nonlinear. As a result, u(t) is considerably different from f(t) in the aspect of wave packet shape. Specifically, the frequency components with high and low phase velocities will appear in the forepart and the rear of the wave packets, respectively. Intuitively, the waveform will be stretched in the time domain after the excitation signal propagates a certain distance in the plate.

### 2.2. Dispersion Compensation Principle

Equation (1) presents that a received guided wave signal is obtained with the propagation distance *d*, as shown in Figure 2. In the forward propagation direction from the origin of the coordinate, the signal is stretched continuously in time t and space x. The same dispersion phenomenon occurs in the backward propagation direction. Thus, in the negative propagation direction, the stretched wave packets converge towards the original excited wave before passing through the x=0 line.

In the distance domain, the compensation waveform H(ϖ) of the excitation signal can be obtained by substituting t=0 into Equation (1). The compensated waveform is as follows:(3)$$\begin{array}{l} H(x)=u(-x,0)=\int_{-\infty }^{+\infty }G(\varpi )e^{ik(\varpi )x}e^{i\varpi 0}\\ \;\;\;\;\;\;\;\;\;\;\;\;\;\;\;\;\;\;\;\;\;\;\;\;\;\;\;\;=\int_{-\infty }^{+\infty }G(\varpi )e^{ik(\varpi )x}d\varpi \end{array}$$
where G(ϖ) is the Fourier transformation of the received signal g(t).

Equation (3) is difficult to implement directly, because the variables x and k hold as a Fourier transform pair [19]. Thus, the variable ω can be transformed into the wavenumber k by defining group velocity as $d\varpi =c_{g}(\varpi )dk$, where $c_{g}(\varpi )$ represents the group velocity. Thus, Equation (3) can be interpreted in the following form:(4)$$H(x)=\int_{-\infty }^{+\infty }G(\varpi )e^{ik(\varpi )x}c_{g}(\varpi )dk=\int_{-\infty }^{+\infty }G(k)e^{ik(\varpi )x}dk$$
In Equation (4), H(x) is in the form of an invers e Fourier transformation of G(k(ϖ)). k(ϖ) is a nonlinear function of ϖ, causing the value of function G(k(ϖ)) to appear unequally spaced. To apply the IFFT algorithm (Inverse Fast Fourier Transform), the dispersion relationship between k(ϖ) and ϖ must be used to interpolate U(ϖ(k)) to U(ϖ(k)).

## 3. Principle of Crack Location

### 3.1. Diffraction Principle

Reflection and diffraction are two types of fundamental phenomena when an acoustic wave encounters discontinuous material. Reflection waves are utilized by numerous scientific studies to localize hole-like damages, whereas diffraction waves from crack endpoints can be used to localize the crack. Thus, a damage imaging method based on diffraction wave theory is proposed in the present study.

When the obstacle size is considerably smaller than the wavelength of the sound wave, the sound wave continues to propagate, similar to the situation without the obstacle. On the contrary, when the obstacle size is considerably larger than the wavelength of the sound wave, the sound wave is mainly reflected, and the area behind the obstacle remains silent. Between these situations, if the obstacle length is similar to the wavelength, then the sound wave diffracts around the obstacle and propagates to the other side of the obstacle, as shown in Figure 3.

Any vibrating point in the wave field can be regarded as a sub-sound source. When the wave front encounters a fatigue crack, the sub-sound source in the crack endpoint propagates to the region behind the crack. Therefore, if a sensor array is mounted in the region behind the crack, then the diffraction wave signals are received, as shown in Figure 4a. Two diffraction wave packets from the crack endpoints are recorded by the sensor array. The received signal of each sensor contains two types of wave packets: S0 and A0 mode. However, the amplitude of S0 mode is considerably small, as shown in Figure 4b so that only the A0 wave packet is used to localize the crack. Affected by the dispersion characteristic, the diffraction wave packets expand in the time domain. Consequently, the crack imaging result based on the signals presents a reasonably low resolution. The time–distance mapping method can narrow down the stretched wave packets, as shown in Figure 4c. Therefore, the overlapped wave packets can be separated to individual wave packets. The compensated signals are beneficial for improving the imaging resolution.

### 3.2. Crack Localization Algorithm

The delay-and-sum algorithm is a widely used imaging method and often based on the time domain signals. Different delay times are initially executed to signals of different channels, as shown in Figure 5a. Then, the residual signals in the time domain are summed, and the maximum value of the summed signal is regarded as the energy of the target point corresponding to the delay times. However, the wave packets of different channels have different lengths due to the dispersion characteristic. Thus, the peak value of different channels in Figure 5a will not be summed at the same time. Actually, different delay times, as shown in Figure 5b, will ensure the peak values are summed simultaneously. The summed results are shown in Figure 5c. This condition is why the conventional delay-and-sum algorithm has a certain localization error.

In this study, the time domain signals are initially mapped to the distance domain via the dispersion compensation algorithm. Then, the residual signals in time domain are obtained through Hilbert transform. Subsequently, the signal in the distance domain is compensated by half the length of the wave packet. On the basis of the wave propagation principle, the actual propagation distance of the wave is from the initial position of the excitation wave to the arrival point of the compensated wave (or from the excitation wave peak to the compensated wave peak), as shown in Figure 6. Therefore, the half-wave packet must be compensated in the distance domain signal. Finally, the energy of each point is obtained by an appropriate shifting rule.

For a sensor pair, the exciter is located at coordinates $\left(x_{i},y_{i}\right)$, and the receiver is located at coordinates $\left(x_{j},y_{j}\right)$. For a point (x,y) in the target field, the propagation distance of a wave, from the exciter to the point and then to the receiver, is as follows:(5)$$d_{xy}=\sqrt{{\left(x_{i}-x\right)}^{2}+{\left(y_{i}-y\right)}^{2}}+\sqrt{{\left(x_{j}-x\right)}^{2}+{\left(y_{j}-y\right)}^{2}}$$

The energy of the point is defined by the relationship between the distance and the enveloped amplitude, as shown in Figure 7a. Therefore, the points with the same distance $d_{xy}$ to the sensor pair have the same energy. It is known that the points with the same distance to two focus points form an ellipse. As a result, the ellipse energy field is established with the sensor pair as the focus points, as shown in Figure 7b. Moreover, the ellipse appears as an elliptical band with a certain width, and the width of the ellipse is relative to the length of the wave packet. Thanks to the dispersion compensation, the wave packet is compressed to a considerably small one, which ensures the elliptical band is reasonable narrow.

Then, all energy values for each sensor pair are summed up to yield the average energy at (x,y) [20].
(6)$$A(x,y)=\left[\frac{1}{N}\sum_{i=1}^{N}c_{i}(d_{xy})\right]$$
where $c_{i}(d_{xy})$ is the relationship between the distance and the amplitude of the enveloped curve.

### 3.3. Performance Analysis

The basic principle of the crack localization algorithm is the diffraction phenomenon. Thus, the proposed method is invalid if no diffraction waves are received by the receiving sensor. In this situation, the crack is unable to block wave propagation. The region where the crack is localized is called blind region for the proposed method, as shown in Figure 8a. It can be seen that the blind region of a horizontal line array is much smaller than that of a vertical line array, with the same spacing distance between receiving sensors. Therefore, the horizontal line array is suggested in this study.

When the excitation sensor is located in the same horizontal line with the receiving sensors, the area of the blind region is minimized. However, it is more likely that the wave path difference between the top and bottom diffraction waves is the same for each receiving sensor, as shown in Figure 8b. Hence, the excitation sensor was located in the top left corner of the horizontal line array in this study.

It is assumed that the lower endpoint of the crack is localized at (*x*, *y*), and the wave path distance from the excitation sensor to the receiving sensor S1, passing by the endpoint of crack, is denoted by P1. In the same manner, the wave path distance for S2 and S3 is denoted by P2 and P3 respectively. If each sensor channel is affected by the same disturbance factors, which cause the propagation distance to change, the distance of each wave path is P1 + ΔP, P2 + ΔP, and P3 + ΔP. As a result, the localization result of the proposed method is denoted by (*x*’, *y*’). Then, the localization error for the actual crack endpoint (x, y) is

(7)$$E(x,y)=\sqrt{{\left(x-x^{\prime }\right)}^{2}+{\left(y-y^{\prime }\right)}^{2}}$$

There are mainly three regions in the localization error diagram: the left side, the middle, and the right side. The localization error enlarges from the left side to the right side gradually, except in the region near the excitation sensor. The offset distance (denoted by D) from the horizontal line array affects the performance of the proposed method, as shown in Figure 9b,c. With the increase of D, the localization errors of the central and right-side region enlarge. Therefore, the offset distance was set to be 10mm in this study.

## 4. Simulation and Experimental Results

### 4.1. Simulation Model

A simulation was conducted in ABAQUS CAE^TM^ application. A simulation model containing a long thickness-through crack was developed with dimensions of 1250 mm × 1250 mm × 2 mm, and the crack length was 100 mm. The coordinates of upper and lower endpoints were (625, 675) and (625, 575), respectively. Material parameters of the simulation model are listed in Table 1.

There are two types of simulation solvers for dynamic problems—explicit and implicit. Compared with the implicit solution, the explicit solution is more economical and accurate for wave propagation simulations. Furthermore, there is no convergence problem in the explicit solution, because it does not need iteration. Although piezoelectric elements are not available in ABAQUS CAE^TM^ explicit, and one needs to apply an equivalent load instead of a voltage as actuation loading, the explicit procedure is strongly recommended by Soorgee after comprehensive comparison with the implicit method [21].

In the numerical simulation, the size of the grid affects the final calculation result. The smaller the grid size is, the more accurate the result will be. However, with the decrease of the grid size, the computational consumption increases exponentially. On the contrary, a larger size might cause considerable simulation error. Therefore, element size is usually limited to one-tenth of the wavelength. In the end, the grid size in this study is set as 1 mm. According to the Nyquist’s theory, the acquisition frequency must be at least twice the maximum frequency component. The central frequency of the excitation signal was 150 kHz, and the acquisition frequency was 2.5 MHz in the experiment. Thus, the time step in the simulation was set to be 0.4 μs in accordance with the experiment.

An actuator and three sensors were mounted using coupling agent on the surface of the plate-like model, as shown in Figure 10a. The positions of the excitation sensor and the receiving sensors are indicated by red points, and “×” represents the endpoint of a crack. To avoid the influence of reflected waves from the model boundaries, the elements of the model boundaries were set as infinite element type (CIN3D8). Figure 10b shows a simulation displacement field of wave motion, in which the reflection, diffraction, and direction arrival waves can be easily found.

### 4.2. Simulation Result

A five-cycle sinusoidal signal modulated with Hanning window was selected as the excitation signal, as shown in Figure 11a.
(8)$$y=Asin\left(2\pi f_{c}\right)\times \left(1-cos\left(2\pi f_{c}/N\right)\right)$$
where *A* is the amplitude of the sinusoidal signal, *N* denotes the cycle number of the modulated signal, and f_c_ represents the time serial ranging from zero to *N*.

The dispersion compensation was applied to the excitation signal, mapping the time domain excitation signal to the distance domain, as shown in Figure 11b. Then, the value of half the wave packet length was calculated, which can be used for the compensation of the distance signal.

The guided wave transmitted to the crack endpoints, at which the diffraction waves were emitted. Then, the diffraction waves continued to propagate and were recorded by the sensors. Figure 12a,c,e shows the time domain signals of the sensors. Figure 12b,d,f represents the compensation signals. The corresponding diffraction wave signals generally overlapped because the lengths of the wave paths from the upper and lower endpoints to the sensors were similar. Moreover, the diffraction wave continued to be stretched at the base of the original wave, which worsened the overlapping phenomenon. Thus, dispersion compensation was implemented to the original time domain signals to ease the situation of overlapping waves, as shown in Figure 12b,d,e.

In the dispersion compensation algorithm, the theoretical dispersive curve of the A0 mode was used. Therefore, the wave packets of the A0 mode were compensated, whereas the wave packets of the S0 mode were further dispersive. As a result, tiny wave packets were observed after separating the two wave packets of the A0 mode in the compensated signals. Although the wave packets of the S0 mode were deformed severely, they did not affect the localization result due to the low amplitude of the S0 mode wave.

The half-wave packet compensation was performed to compensate the distance domain signal of the A0 mode. The final imaging results are shown in Figure 13. After compensation, the distances of the first and second wave packet peaks became similar to the actual distance of the wave paths, as shown in Table 2. Localizing the crack with a reasonably high accuracy was possible by using the dispersion and half-wave packet compensation signals.

## 5. Experimental Verification

In an actual experiment, the imaging result can be affected by noise, sensor inconsistency, and other factors. An experimental verification was conducted on a plate with the same parameters as the simulation model to validate the performance of the proposed method. The experimental setup consisted of a Tektronix AFG1022 function/arbitrary waveform generator, a DS2−8B data collecting instrument, four RS−2A sensors, and Smart AE charge amplifiers.

The frequency response band of the RS-2A sensor was 50 kHz–400 kHz with a central frequency of 150 kHz. The dimension of the RS-2A sensor was 18.8 mm in diameter and 15 mm in height. The relative dielectric constant of the RS-2A sensor in the z direction was far larger than that in the x and y directions, and the relative dielectric constants in the x and y directions were approximately equal to 0. The sensors were adhered to the surface of the specimen by coupling agent, among which the sensor on the left was considered the exciter and the remaining sensors are the receivers. The excitation signal was a narrowband tone burst signal with an amplitude of 10 V peak-peak voltage and a center frequency of 150 kHz. The acquisition rate of the data-collecting instrument was 2.5 MHz. Figure 14 shows the connection relationship of the experimental setup.

The signals obtained in the experiment, as shown in Figure 15, were similar to those in the simulation, mainly due to the following reasons: First, the A0 and S0 mode were found in signals of each channel, and only the A0 mode waves were compensated. Second, the amplitude of the A0 and S0 mode waves was different from the simulation signals, especially that of the S0 mode waves, due to the sensor response. Moreover, reflection waves from the boundaries existed in the experiment. Thus, additional wave packets were evident in the experimental signals.

The compensated signals were used to localize the crack. Each sensor pair produces an energy field, and the average of all energy fields was obtained to present the imaging result. The maximum values represent the endpoints of the target crack, and the symbol “+” denotes the actual crack endpoints, as shown in Figure 16a. In comparison, Figure 16b presents the imaging result when the half-wave compensation was not involved. The localized endpoints were far from the actual endpoints. Figure 16c shows the image based on the conventional delay-and-sum algorithm, in which the dispersion effect was not compensated. The endpoints were blurred and combined.

To evaluate the positioning result quantitatively, the ratio of the positioning error in the direction of a certain coordinate axis is as follows:(9)$$E_{x}=\left|\frac{X_{T}-X_{A}}{X_{A}}\right|\times 100\%$$
where $X_{T}$ is the localization value in the direction of a coordinate axis and $X_{A}$ denotes the actual value.

The proposed method had a higher localization accuracy than the delay-and-sum method without half-wave compensation, as presented in Table 3. The localizing error in the *x* coordinate axis was larger than that in the *y* coordinate axis. The error ratio values indicate that the sensor array is more sensitive to the *X*-axis than the *Y*-axis and is susceptible to factors, such as compensation deviation.

## 6. Conclusions

In this study, a crack localization method based on diffraction wave theory is proposed. The original signal is mapped to the distance domain through the time–distance dispersion compensation method. After half-wave packet compensation, a high-resolution damage image is obtained by the summation of the energy field. Simulation and experimental results show that the proposed method can locate two crack endpoints with reasonable accuracy. Conclusions can be drawn as follows:The dispersion compensation of ultrasonic guided wave signal can effectively reduce the length and overlap of wave packets. The half-wave compensation can improve the resolution of the delay-and-sum imaging method.On the basis of the principle of acoustic diffraction, a line sensor array can localize the crack endpoints.The offset distance between the excitation sensor and the receiving sensor is related to the location error of crack damage.


The effectiveness of the crack localization method was verified by simulation and in an experiment, but the actual engineering application is faced with a complex environment, such as high noise level and wave attenuation. In addition, the shape of the sensor array affects the positioning results. Future studies will be gradually conducted to reduce the influence of these factors.

## Figures and Tables

**Figure 1 sensors-19-01951-f001:**
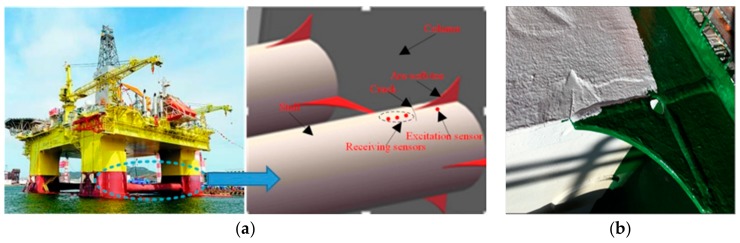
Offshore platforms arc-soft-toe bracket. (**a**) Pipe with a large radius. (**b**) plate-like structure.

**Figure 2 sensors-19-01951-f002:**
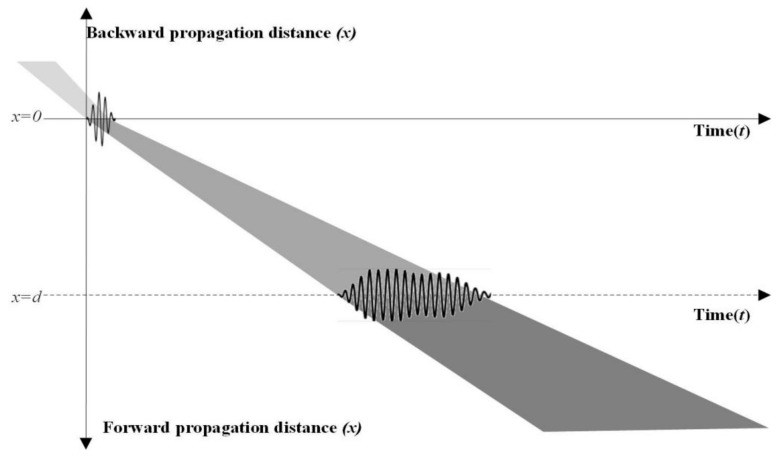
Dispersion of guided wave in the time and space domain.

**Figure 3 sensors-19-01951-f003:**
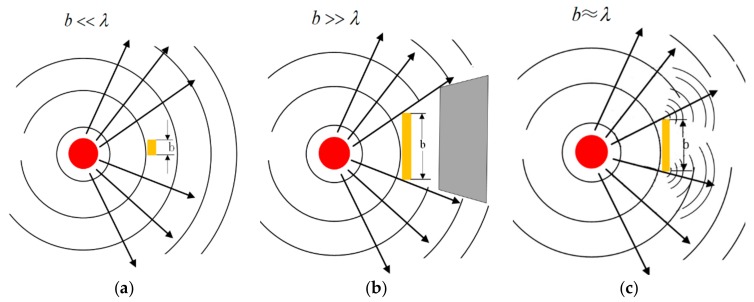
Huygens principle of acoustic propagation patterns. (**a**) The size of the obstacle is much smaller than the diffraction wavelength. (**b**) The size of the obstacle is much larger than the diffraction wavelength. (**c**) The size of the obstacle is approximately equal to the diffraction wavelength.

**Figure 4 sensors-19-01951-f004:**
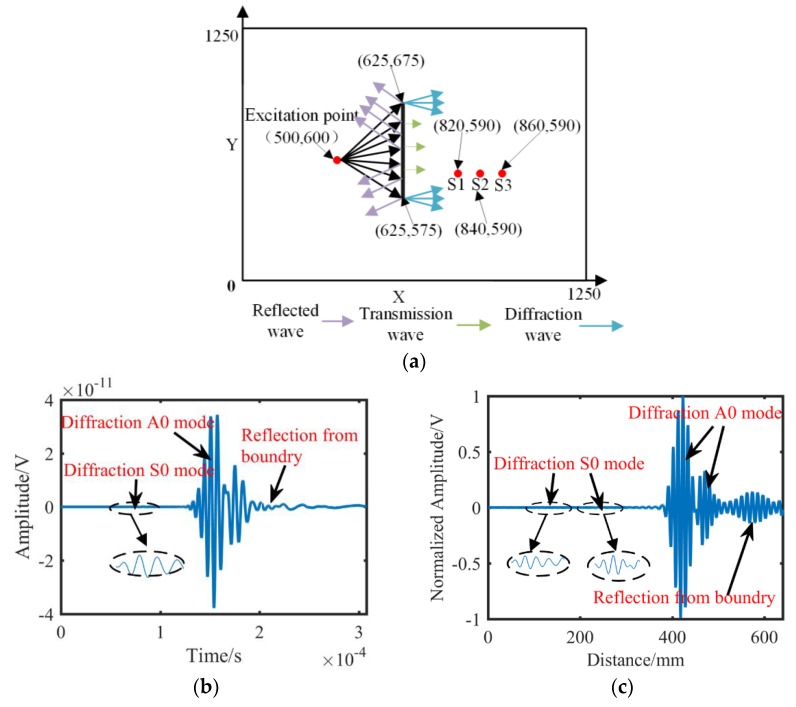
Detection principle and signal waveforms. (**a**) Schematic diagram of detection. (**b**) Time domain signal. (**c**) Distance domain signal.

**Figure 5 sensors-19-01951-f005:**
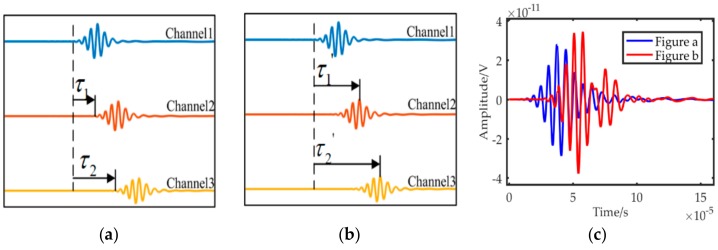
The time-and-delay algorithm. (**a**) Delay times (**b**) residual signals (**c**) summation results.

**Figure 6 sensors-19-01951-f006:**
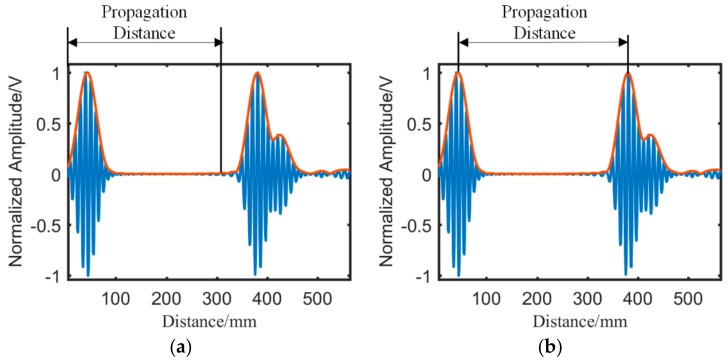
Diagram of waveform propagation distance. (**a**) Between starting points. (**b**) Between peaks.

**Figure 7 sensors-19-01951-f007:**
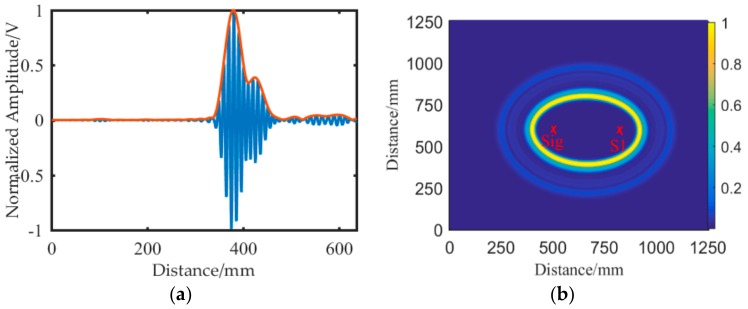
The signal and corresponding energy field for a sensor pair. (**a**) Signal in distance domain (**b**) The ellipse-like energy field.

**Figure 8 sensors-19-01951-f008:**
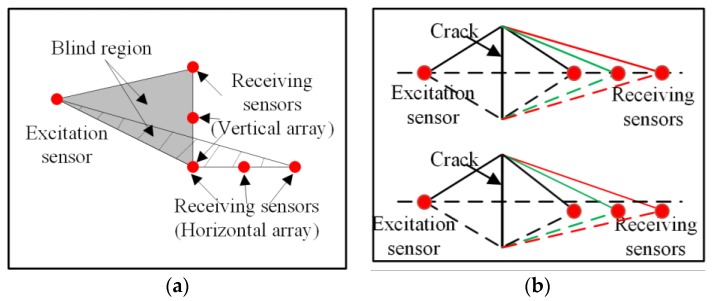
Schematic diagram of the proposed method. (**a**) Sensor array type and blind region. (**b**) The wave path difference.

**Figure 9 sensors-19-01951-f009:**
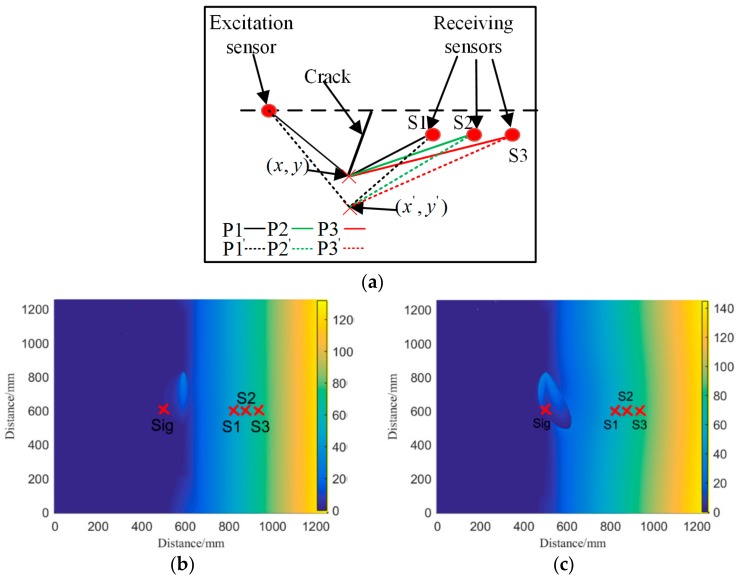
The localization error. (**a**) The change of wave path by adding noise. (**b**) The localization error diagram (ΔP = 3 mm, D = 10 mm). (**c**) The localization error diagram (ΔP = 3 mm, D = 50 mm).

**Figure 10 sensors-19-01951-f010:**
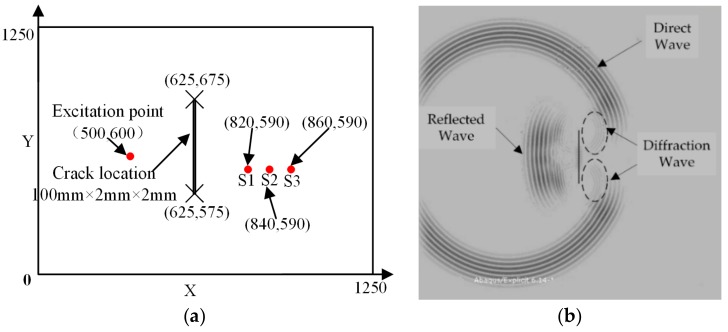
Schematic diagram of guided wave signal fluctuation field and sensor arrangement. (**a**) Simulation model. (**b**) Displacement field of wave motion.

**Figure 11 sensors-19-01951-f011:**
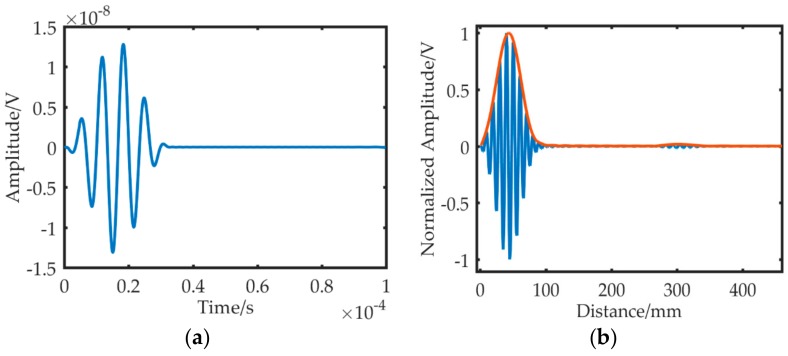
Time and distance domain waveform of excitation signal. (**a**) Time domain signal. (**b**) Distance domain signal.

**Figure 12 sensors-19-01951-f012:**
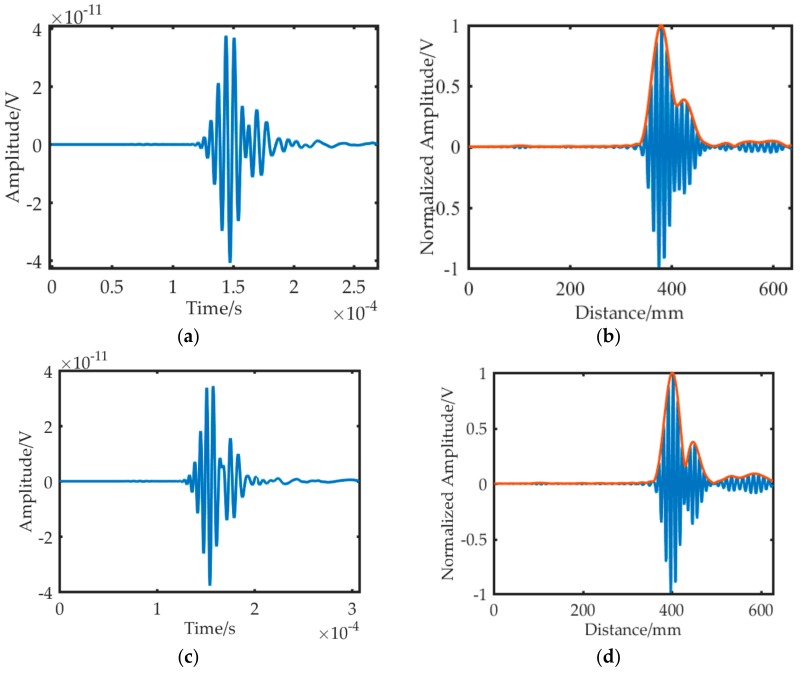
Time and distance domain signals in simulation. (**a**) Time-domain signal for S1. (**b**) Distance-domain signal for S1. (**c**) Time-domain signal for S2. (**d**) Distance-domain signal for S2. (**e**) Time-domain signal for S3. (**f**) Distance-domain signal for S3.

**Figure 13 sensors-19-01951-f013:**
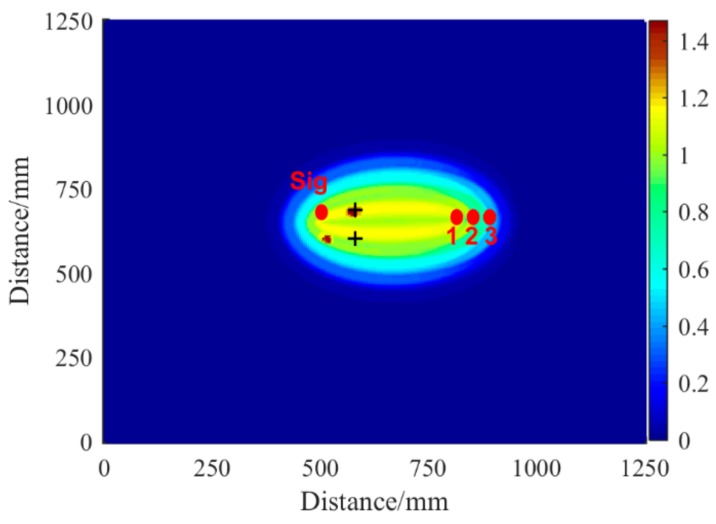
Schematic diagram of delayed superposition imaging in simulation.

**Figure 14 sensors-19-01951-f014:**
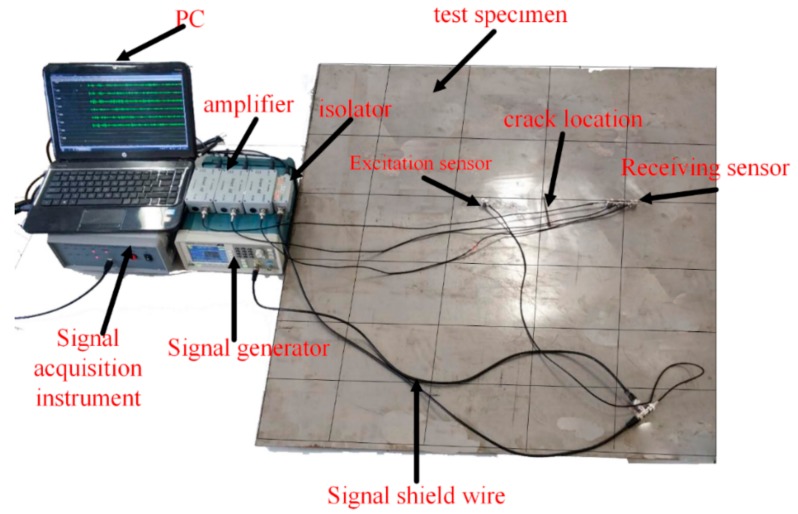
Schematic diagram of the experimental setup.

**Figure 15 sensors-19-01951-f015:**
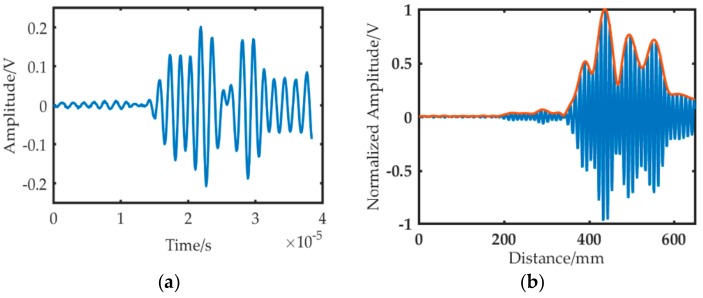
Time and distance domain signals in the experiment. (**a**) Time-domain signal for S1. (**b**) Distance-domain signal for S1. (**c**) Time-domain signal for S2. (**d**) Distance-domain signal for S2. (**e**) Time-domain signal for S3. (**f**) Distance-domain signal for S3.

**Figure 16 sensors-19-01951-f016:**
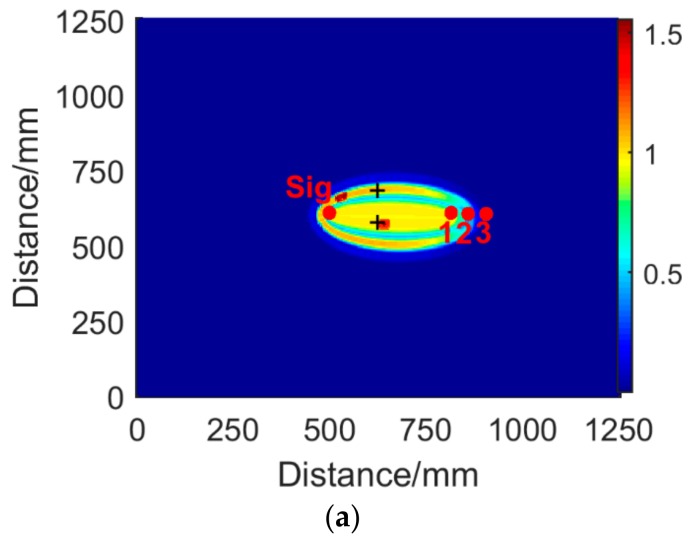
The results of crack imaging. (**a**) Image of the proposed method. (**b**) Image without half-wave compensation. (**c**) Image of conventional delay-and-sum method.

**Table 1 sensors-19-01951-t001:** Material parameters.

Material	Density (kg/m^3^)	Elastic Modulus (Pa)	Poisson’s Ratio
Q235	7850	2.1 × 10^11^	0.3

**Table 2 sensors-19-01951-t002:** Actual wave path length and compensation results.

	Sig-S1	Sig-S2	Sig-S3
Wave path of the lower endpoint (mm)	323	343	363
Peak of first wave packet (mm)	322	344	364
Wave path of the upper endpoint (mm)	358	376	395
Peak of second wave packet (mm)	356	377	398

**Table 3 sensors-19-01951-t003:** The localization results of endpoints.

Method	Localization Results			
	Upper Endpoint	Error Ratio	Lower Endpoint	Error Ratio
The proposed	(530,660)	X:15%; Y:2%	(640,570)	X:2.4%, Y:0.87%
Without half-wave compensation	(475,630)	X:30%, Y:6%	(525,555)	X:16%, Y:3.5%

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
