# Peer review of "High-Resolution Crack Localization Approach Based on Diffraction Wave"

_sensors, 2019, doi:10.3390/s19081951_

Round 1

Reviewer 1 Report

Review of Article: High resolution crack localization approach based on diffraction wave by Wailei Mu et al.

The article discusses the use of a half-wave compensation method based on the delay and sum imaging algorithm for damage localization. The article in general is well written and clear to follow. The authors have given a good insight into the methodology that they have followed. In addition, the study makes use of two scientific methods to prove their algorithms. However, there are improvements that the article requires for it to be worthy of acceptance as a journal publication. As a reviewer I will divide my comments into two categories: (i) minor changes which are easily fixed; and (ii) more in-depth changes that require more process thought.

Minor comments:

1.     Abstract, first line: approach for detection and monitoring… of what?

2.     Introduction: first line: you discuss that this study is being focused for offshore platforms, however the authors ignore one of the most common problems in offshore platforms is corrosion.

3.     Chinese characters on the top of each page.

4.     Case 2.1: Cause of dispersion… Your considering guided waves in plate like structures. What part of the offshore oil rigs are you considering?

5.     Section 3.2: Line: Thus, the previous summation….. of each channel is.. Please rephrase, its unclear..

6.     ABAQUS should be ABAQUS CAETM

7.     Table 3, make sure its on the same page and not split, it might just be my PDF version on my IPAD.

Major comments:

1.     Equation 1, considers a central frequency of a signal, such as a Hanning Window, however cracks emit signals which are broadband in nature. I understand that your study is based on a pitch catch scenario, but then the question comes how do you relate this type of setup to a real-world case of an oil rig.

2.     The authors make the statement that the Cp (phase velocity) is often non-linear, however, they don’t explain how they deal with this for a real world application like an oil rig? Maybe it would be better to re-write the manuscript in terms of a fundamental work.

3.     The study makes use of a pitch-catch case scenario where the “crack” is exactly at 90 degrees to the path of the signal. What would happen if your crack is oriented at 0 degrees or 45 degrees. Since it’s a pitch catch scenario the authors don’t know at priori what the orientation of the crack to the signal path.

4.     A crack is not an opening. A crack, specially a fatigue crack is in many cases closed, how would this affect the ability of your algorithm to catch the location of damage.

5.     The authors make use of ABAQUS to model their setup, however, no explanation is given on how the model is developed. Mesh size, time step, Implicit vs. Explicit, how is their actuator sensor modeled?

6.     Figure 3a needs more detail information of the geometry, crack size, distances from the actuator to sensor, etc.

7.     Figure 3b, should highlight the So wave packet better.

8.     No information of piezoelectric material properties. If the author modeled the actuation signal as a point load vs. time, this would ignore some of the piezoelectric coupled effects.

9.     The authors seemed to have setup the actuator and sensors at such a distance from each other that creates two clear packets (So and Ao). However, reality would have temperature effects, boundary conditions, loading conditions, reflections all making these wave packages on top of each other and much noisier than their very nice laboratory test. How are the authors taking care of this?

10.  What would happen if the sensor location changed? How would their algorithm react on the ability of determining the crack fronts/tips?

General comments:

            In its current form it would be difficult to recommend publication of this manuscript. I think it’s more like a conference paper than a journal manuscript. I think it’s a nice piece of work but it requires a more in-depth discussion and insight in order to make it worthy of a journal publication.

Author Response

Minor comments:

1.     Abstract, first line: approach for detection and monitoring… of what?

Response: Thanks for your advice. We have changed the sentence ‘The delay-and-sum imaging algorithm is a promising localization approach for detection and monitoring’ to ‘The delay-and-sum imaging algorithm is a promising crack localization approach for crack detection and monitoring of key structural region.’ in the abstract

2.        Introduction: first line: you discuss that this study is being focused for offshore platforms, however the authors ignore one of the most common problems in offshore platforms is corrosion.

This is a good suggestion, we have amend the first paragraph of introduce to” During service, offshore platforms are affected by wind, wave, sea current and other alternating loads, which can lead to fatigue cracks in key welding region, such as welds of arc-soft-toe bracket (ASTB) joints. Moreover, Corrosion and iceberg impact also produce fatigue cracks in critical areas, even though the welding quality is good. The crack appears because the bracket toe is not soft enough to release the stress concentration, therefore, cracks often appear at the end of the toe of the elbow and the direction is approximately perpendicular to the weld”

3.        Chinese characters on the top of each page.

Response: I am sorry that the problem was neglected. We have deleted Chinese characters in the manuscript

4.        Case 2.1: Cause of dispersion… Your considering guided waves in plate like structures. What part of the offshore oil rigs are you considering?

Thank you very much for your suggestion. We have explained as follows:

Generally, the structural elements connected by ASTB joints in the offshore platform are plate-like components or pipes with large radius, as shown in Fig.1. In a relatively small region, the pipe could be regarded as a plate-like component approximately.

Figure 1 .Offshore platforms arc-soft-toe bracket. (a) Pipe with a large radius. (b) plate-like structural

5.        Section 3.2: Line: Thus, the previous summation….. of each channel is.. Please rephrase, its unclear.

Response: I'm sorry we didn't explain this clearly, in order to make it clearer, we have modified in the first paragraph of section 3.2. on Page 6 to “However, the wave packets of different channels have different lengths due to the dispersion characteristic. Thus, the peak value of different channels in Fig. 4(a) will not be summed at the same time. Actually, different delay times, as shown in Fig. 4(b), will ensure the peak values summed simultaneously. The summed results are shown in Fig. 4(c).”

6.        ABAQUS should be ABAQUS CAETM

Thanks for your advice we have revised in the manuscript

7.        Table 3, make sure its on the same page and not split, it might just be my PDF version on my IPAD.

Thanks for your advice we have revised in the manuscript in Table 3 on Page 15

Major comments:

1.        Equation 1, considers a central frequency of a signal, such as a Hanning Window, however cracks emit signals which are broadband in nature. I understand that your study is based on a pitch catch scenario, but then the question comes how do you relate this type of setup to a real-world case of an oil rig.

Response: Thank you very much for your suggestion.

Actually, we want to monitor the key region of arc-soft-toe joints. If it is permitted by the ship classification society, we want to install the excitation-receiving sensor on the transverse bracing surface in the future, as shown in Fig. 2. It is really a challenging task. So, we have to do some experiments in the lab, and expecting that the experiments could confirm well with the real application.

Indeed, the frequency band of acoustic emission signal from the crack is broadband, the passive localization method is proposed in our previous study. In this paper, we use an active localization method. Therefore, in the pitch-catch scenario, the narrowband signal modulated by Hanning is used as excitation to realize damage detection in key areas of offshore platforms.

Figure 2 Schematic diagram of sensor layout

2.        The authors make the statement that the Cp (phase velocity) is often non-linear, however, they don’t explain how they deal with this for a real world application like an oil rig? Maybe it would be better to rewrite the manuscript in terms of a fundamental work.

Response: Thank you for your suggestion. According to the dispersion characteristic curve, has the dispersion characteristic in the plate structure. Generally, the structural elements connected by arc-soft-toe bracket (ASTB) joints in the offshore platform are plate-like components or pipes with large radius. In a relatively small region, the pipe could be regarded as a plate-like component approximately (revised in section 2.1). After that, the nonlinear relationship between phase velocityand frequencyis used in the dispersion compensation to compress the dispersive wave packet.

3.        The study makes use of a pitch-catch case scenario where the “crack” is exactly at 90 degrees to the path of the signal. What would happen if your crack is oriented at 0 degrees or 45 degrees. Since it’s a pitch catch scenario the authors don’t know at priori what the orientation of the crack to the signal path.

Response: I'm sorry we didn't explain this clearly. Usually, the crack appears when the bracket toe is not soft enough to release the stress concentration. Therefore, cracks often appear at the end of the toe of the elbow and the direction is approximately perpendicular to the weld. Revised in section 1, the first paragraph. Furthermore, the proposed method is capable of localizing cracks with any degrees, except the blind region. We add a new subsection (Section 3.3 on Page 7) to explain the performance of the proposed method following your advice, which could give a more comprehensive explanation.

4.        A crack is not an opening. A crack, specially a fatigue crack is in many cases closed, how would this affect the ability of your algorithm to catch the location of damage.

Response: That's a good advice for my manuscript. In the actual detection environment, open crack are much easier to detect and characterize than closed crack, and the close fatigue crack do affect the ability of localization algorithm. The crack at the axial toe of the axial plate of the offshore platform is usually caused by the concentration of stress (alternating stress). When the stress reaches the fatigue limit at the base plate, the fatigue crack will be generated and the stress will be released. Thus, the aiming crack in this study is often open. We are sorry that we did not supply enough background information in the original manuscript at first.

In many cases, the crack is closed, and the detection of close crack is a challenging task we would like to try. The nonlinear guided wave is a promising method, which we would like to use. Thanks for your kindly advice.

5.        The authors make use of ABAQUS to model their setup, however, no explanation is given on how the model is developed. Mesh size, time step, Implicit vs. Explicit, how is their actuator sensor modeled?

Response: This is a good suggestion, in the second paragraph of section 4.1, we have made the following detailed description of parameter setting.

There are two types of simulation solver for dynamic problem, explicit and implicit. Compared with implicit solution, explicit solution is more economical and accurate for wave propagation simulation. Furthermore, there is no convergence problem in explicit solution, because it does not need iteration. Although piezoelectric element are not available in ABAQUS CAETM explicit, and one needs to apply an equivalent load instead of a voltage as actuation loading, the explicit procedure is strongly recommended by Soorgee, M.H. after comprehensive comparison with implicit method.

In the numerical simulation, the size of the grid will affect the final calculation result. The smaller the grid size is, the more accurate the result will be. However, with the decrease of the grid size, the computational consumption will increase exponentially. On the contrary, the larger size might cause considerable simulation error. Therefore, element size is usually limit into one-tenth of the wavelength. The grid size in this study is set as 1 mm. According to the Nyquist’s theory, the acquisition frequency must be twice more than the maximum frequency component at least. The central frequency of the exciting signal is 150 kHz, and the acquisition frequency is 2.5 MHz in the following experiment. Thus, the time step in the simulation is set to be 0.4 μs in accordance with the experiment.

6.        Figure 3a needs more detail information of the geometry, crack size, distances from the actuator to sensor, etc.

Thanks for your advice we have added the detail information in Figure 4(a) on Page 5

7.        Figure 3b, should highlight the So wave packet better.

Response: Thank you for your reminding, we have highlighted the S0 wave packet in Figure. 4(b) (c) on Page 5

8.        No information of piezoelectric material properties. If the author modeled the actuation signal as a point load vs. time, this would ignore some of the piezoelectric coupled effects.

Response: I am sorry that the problem was neglected, we have supplemented the information of piezoelectric material properties in the first paragraph of section 5.

The frequency response band of RS-2A sensor is 50 kHz-400 kHz with a central frequency of 150 kHz. The dimension of RS-2A sensor is with 18.8 mm in diameter, and 15 mm in height. The relative dielectric constant of RS-2A sensor in the z direction is far larger than that in the x and y directions, and the relative dielectric constants in the x and y directions are approximately equal to 0 (added in section 5). Therefore, only the displacement in the z direction is extracted in the simulation process of ABAQUS CAETM , and the piezoelectric coupling effect in the x and y directions is ignored.

9.        The authors seemed to have setup the actuator and sensors at such a distance from each other that creates two clear packets (So and Ao). However, reality would have temperature effects, boundary conditions, loading conditions, reflections all making these wave packages on top of each other and much noisier than their very nice laboratory test. How are the authors taking care of this?

Response: This is a good suggestion for my manuscript, we will continue to make effort to eliminate the effect of environmental factors.

Two diffraction wave packets from the crack endpoints are recorded by the sensor array. The received signal of each sensor contains two types of wave packets: S0 and A0 mode. However, the amplitude of S0 mode is considerable small, as shown in Fig. 4(b). So that, only the A0 wave packet will be used to localize the crack.(revised in Page 5)

In this situation, the speed of A0 wave packet might be affected by the environmental factors. However, according to Hoon Sohn’s study, environmental variations like temperature change do not affect its performance, since the method detects damages without having to rely on baseline data.[ Sohn H, Dutta D. Temperature Independent Damage Detection in Plates Using Redundant Signal Measurements[J]. Journal of Nondestructive Evaluation, 2011, 30(2):106-116.]

However, boundary conditions and reflection may lead to complete or partial overlap between signal wave packets. In the case of complete overlap, each wave packet could be decompose by matching pursuit or signal decomposition. [Kim H W , Yuan F G . Enhanced Damage Imaging of a Metallic Plate Using Matching Pursuit Algorithm with Multiple Wavepaths[J]. Ultrasonics, 2018.] The signal decomposition (matching puisuit) is what we are planning to do in the future. We are sorry we could not supply the result now, to answer your question.

While, if only partial waveforms are overlapped, the proposed method could localize the crack, because the peak of the wave packet is essential for this method.

10.     What would happen if the sensor location changed? How would their algorithm react on the ability of determining the crack fronts/tips?

Response: Thank you for your suggestion. Following your suggestion, we add a new subsection 3.3, which analysis the performance of the proposed method. The change of sensor location will affect the accuracy, indeed.

Time-lapse superposition imaging technology is the elliptic field drawn with the position of the excitation sensor and the receiving sensor as the focus. When the position of the excitation-receiving sensor is changed, the most intuitive change is that the size of the generated elliptic field will change. When the position of the sensor is changed to increase the size in the y direction, matlab software is used to analyze and it is known that the increase of crack imaging error will lead to the increase of monitoring blind area. When changing the position of the sensor to reduce the size in the y direction, it may lead to the extreme situation that the wave packets receiving the sensor completely overlap. Changing the position of the sensor makes the size change in the x direction, which has little impact on the final imaging results.

Reviewer 2 Report

General remarks:

The reference numbers do not follow to the list in the reference section

The quality of the figures should be improved according to the journal requirements

The numerical model of the analyzed wave propagation case should be explained in details.The results of the crack imaging should be also presented for the numerical model.
The manuscript text and form should be deeply revised according to the journal requirementsIn the reviewer opinion, the influence of the crack size and orientation should be additionally analyzed with the use of the proposed method. Additionally, the localization of the actuator and sensors in the carefully chosen places, relatively close to the crack raises doubts about the application of the method in real crack detection, even in the experimental fatigue tests.
Additional remarks can be found as comments to the attached pdf file. 

Author Response

1.        The assumption of the elliptic shape of the energy field should be better emphasized. How authors control the width of the ellipse?

Response: Thank you for your suggestion, we have described as follows in the above paragraph of equation 6 on Page 7: the points with the same distance  to the sensor pair will have the same energy. It is known that the points with the same distance to two focus points will form an ellipse. As a result, the ellipse energy field is established with the sensor pair as the focus points, as shown in Fig. 7(b). Moreover, the ellipse appears as an elliptical band with a certain width, and the width of the ellipse is relative with the length of wave packet. Thanks to the dispersion compensation, the wave packet is compressed to the considerable small one, which ensure the elliptical band is reasonable narrow.

2.        The global coordinate system should be marked in the Fig 7a if the numbers in brackets mean the position of the particular points. What the red markers mean. In the text, the one actuator and three sensors are described so what means the markers at the tips of crack?

Response: Thank you very much for your advice. The red mark in the figure represents the position of the excitation sensor and the receiving sensor, and now we use ‘×’ represents the endpoints of the crack, for distinguishing with the sensors.

3.        What is presented in Fig. 7b, the displacement field ?

Response: I'm sorry we didn't explain this clearly. We have changed the title to “Displacement field of wave motion” in Figure 10(b) on Page 10 (the number of figure is changed, because new figures are inserted before).

4.        The numerical model of the analyzed wave propagation case should be explained in details. The wave propagation phenomenon is strongly dependent on mesh density. Do the authors the convergence tests of the presented results. What authors mean by actuator and sensors mounted on the surface of the plate. What kind of sensors were introduced in the numerical model? The detailed description of the crack model in the numerical model should also be presented.

Response: This is an essential suggestion for us. In the second paragraph of section 4.1, we have made the following detailed description of parameter setting.

There are two types of simulation solver for dynamic problem, explicit and implicit. Compared with implicit solution, explicit solution is more economical and accurate for wave propagation simulation. Furthermore, there is no convergence problem in explicit solution, because it does not need iteration. Although piezoelectric element are not available in ABAQUS CAETM explicit, and one needs to apply an equivalent load instead of a voltage as actuation loading, the explicit procedure is strongly recommended by Soorgee, M.H. after comprehensive comparison with implicit method.

In the numerical simulation, the size of the grid will affect the final calculation result. The smaller the grid size is, the more accurate the result will be. However, with the decrease of the grid size, the computational consumption will increase exponentially. On the contrary, the larger size might cause considerable simulation error. Therefore, element size is usually limit into one-tenth of the wavelength. The grid size in this study is set as 1 mm. According to the Nyquist’s theory, the acquisition frequency must be twice more than the maximum frequency component at least. The central frequency of the exciting signal is 150kHz, and the acquisition frequency is 2.5 MHz in the following experiment. Thus, the time step in the simulation is set to be 0.4 μs in accordance with the experiment.

The crack is a long thickness-through crack, with the length of 100 mm, and the endpoint coordinates are respectively (625,575) and (625,675).

5.        The orientation of the crack relatively to the actuator and sensors is different than in numerical analysis?

Response: In the Fig. 14, the orientation of the crack looks different with that in numerical model, because of the orientation of photographing. While the location of the sensor, the length and the location of the crack in the experiment are completely consistent with the simulation model.

6.        The results of the crack imaging should be also presented for the numerical results.

Response: I am sorry that the problem was neglected. We have presented the crack imaging for the numerical results in Figure 13 on Page 12.

7.        what kind of sensors were used, the sensors were permanently integrated with the structure or there were some other methods of surface monting (glue, tape, wax...)

Response: The sensor model used in the experiment is RS-2A sensor, which is connected to the surface of the specimen by coupling agent in laboratory environment. While, the piezoelectric wafers might be fixed by AB glue during the detection in key areas of the offshore platform in the future, if it is permitted by ship classification society.

8.        The manuscript text should be revised according to the journal requirements

Response: Thanks for your advice we have revised in the manuscript.

9.        In the reviewer opinion, the influence of the crack size and orientation should be additionally analyzed with the use of the proposed method. Additionally, the localization of the actuator and sensors in the carefully chosen places, relatively close to the crack raises doubts about the application of the method in real crack detection, even in the experimental fatigue tests.

Response: I'm sorry we didn't explain this clearly

Actually, the proposed method is capable of localizing cracks with any degrees, except the blind region. We add a new subsection (Section 3.3 on Page 7) to explain the performance of the proposed method following your advice, which could give a more comprehensive explanation. Moreover, we have discuss the influence of crack endpoint to the localization error. The different sensor array and different offset distance between excitation and receiving sensors are also discussed.

10.     reference mistake

Response: Sorry for this mistake, we have revised in the manuscript

Round 2

Reviewer 2 Report

The reference enumeration has to be corrected in the text. Many “Error! Reference source not found” was found in the text.

The text justification should be applied in a few places.

The mistake in first name and surname was found in the reference [16]  

Author Response

Thank you for your letter and for the reviewers’ comments concerning our manuscript entitled “High-resolution crack localization approach based on diffraction wave”. Those comments are all valuable and very helpful for revising and improving our paper, as well as the important guiding significance to our researches. We have studied comments carefully and have made correction which we hope meet with approval. The main corrections in the paper and the responds to the reviewers’ comments are as following:

Point 1:The reference enumeration has to be corrected in the text. Many “Error! Reference source not found” was found in the text

Response 1: I am sorry for the cross-reference errors, which we didn’t find during the double check. Now, we have revised in the manuscript.

Point 2. The text justification should be applied in a few places

Response 2: After double-check, we have found some places needed to be corrected, and we have made some correction in the manuscript. 

Point 3The mistake in first name and surname was found in the reference [16].

Response 3: I am so sorry to make the silly mistake, we have corrected the full names of the authors on page 17

Thanks for your professional comments again.
